# Inkjet-Printed Multiwalled Carbon Nanotube Dispersion as Wireless Passive Strain Sensor

**DOI:** 10.3390/s24051585

**Published:** 2024-02-29

**Authors:** Abderrahmane Benchirouf, Olfa Kanoun

**Affiliations:** Measurements and Sensor Technology, Reichenhainer Str. 70, 09126 Chemnitz, Germany; benchirouf@ieee.org

**Keywords:** multiwalled carbon nanotubes, inkjet printing, strain sensor, wireless sensing, impedance spectroscopy, inductive coupling

## Abstract

In this study, a multiwalled carbon nanotube (MWCNT) dispersion is used as an ink for a single-nozzle inkjet printing system to produce a planar coil that can be used to determine strain wirelessly. The MWCNT dispersion is non-covalently functionalized by dispersing the CNTs in an anionic surfactant, namely sodium dodecyl sulfate (SDS). The fabrication parameters, such as sonication energy and centrifugation time, are optimized to obtain an aqueous suspension suitable for an inkjet printer. Planar coils with different design parameters are printed on a flexible polyethylene terephthalate (PET) polymer substrate. The design parameters include a different number of windings, inner diameter, outer diameter, and deposited layers. The electrical impedance spectroscopy (EIS) analysis is employed to characterize the printed planar coils, and an equivalent electrical circuit model is derived based on the results. Additionally, the radio frequency identification technique is utilized to wirelessly investigate the read-out mechanism of the printed planar MWCNT coils. The complex impedance of the inductively coupled sensor undergoes a shift under strain, allowing for the monitoring of changes in resonance frequency and bandwidth (i.e., amplitude). The proposed wireless strain sensor exhibits a remarkable gauge factor of 22.5, which is nearly 15 times higher than that of the wireless strain sensors based on conventional metallic strain gauges. The high gauge factor of the proposed sensor suggests its high potential in a wide range of applications, such as structural health monitoring, wearable devices, and soft robotics.

## 1. Introduction

Conventional strain sensors, also known as metallic strain gauges, can measure strains only on the structural surface; therefore, wires are needed to acquire data. The need for wires will increase the complexity of the system if high numbers of strain sensors are used on a large surface area, as in structural health monitoring or airplane wings …etc.; in any case, it cannot be implemented in rotational parts or buried into structures. Wireless sensors, on the other side, give the possibility of eliminating the need for wires and thus overcome the pre-mentioned limitations of the traditional strain sensors. Several industrial products are aimed at realizing wireless strain sensors based on metal gauges that utilize the principle of piezoresistivity to gauge strain-induced changes in resistance [1,2]. These sensors offer several advantages, including reduced installation and maintenance costs, enhanced flexibility, real-time data acquisition, continuous monitoring, improved safety, and reduced cabling hazards. However, potential interference, limited communication range, power consumption, and higher initial costs remain as limitations. Carbon nanotubes (CNTs) are a promising nanomaterial for strain sensors because of their outstanding mechanical and electrical properties; also, it possesses a high aspect ratio and very low density [3,4,5,6]. Enormous studies showed the feasibility of using CNTs-based thin films for strain sensing, their unique physical mechanism for measuring strain is based on how electrical resistance changes in response to applied mechanical stress, allowing for accurate and reliable measurements [7,8,9,10]. Generally speaking, when the CNT thin films are under applied strain force, an increase in the resistance of the film is observed. This change in the electrical resistance is due to an altering of the tunneling distance between the neighboring CNTs that form the conduction paths [11,12,13,14]. Though thin film-based CNTs showed high potentials for strain sensing applications, temperature effect, stability, and linearity are some unsolved problems [15,16,17]. Thin films composed of CNTs are made using a variety of deposition methods, such as drop-casting, spin-coating, spray-coating, dielectrophoresis [18], and screen printing [19], each of which has advantages and disadvantages that are unique [20,21,22]. These deposition methods enable tailored CNT dispersion deposition for an array of applications by providing fine control over the film thickness and uniformity. Among others, layer-by-layer (LbL) deposition is a versatile and controllable method for the synthesis of thin-film CNTs. This technique involves the sequential assembly of CNTs layer by layer, resulting in well-defined structures with individualized characteristics. The CNTs are typically dispersed in a solution with a countercharged poly-electrolyte and other charged characteristics. The substrate is alternately immersed in the CNT dispersion and rinsed with a solvent or buffer solution between each deposition step. During this process, oppositely charged species are absorbed onto the substrate surface, creating a multilayered film with a well-defined structure [23,24,25]. Previously, Loh et al. fabricated a coil structure on a flexible substrate using the layer-by-layer deposition technique [26] by the self-assembly of the multi-bilayers ${\left(SWCNT-PSS/PVA\right)}_{n}$ (*n* denoted the number of the deposited layer) process to coil pattern. The patterned structure was used as a pH sensor as well as a passive wireless strain sensor for localized strain determination. Besides the low strain sensitivity, which was about 1.8, the deposition process is complex and time- and material-consuming. In [27], utilizes multiwall carbon nanotubes (MWCNTs) and polypropylene (PP) composites for flexibility and affordability as thin film and could achieve a strain sensitivity of 4.5. The proposed sensor employs a system identification approach to minimize sensor errors and ensure accurate strain detection, which has an ultra-low-power consumption of 0.8705 mW using LoRa communication, and in [28], the MWCNT/epoxy films exhibit a strain-dependent resonant frequency shift, allowing for wireless strain detection by measuring this shift. Yet, inkjet printing is a suitable deposition method for sensors-based CNT thin films that might overcome the disadvantages of the layer-by-layer technique [29,30,31,32,33,34]. The inkjet printing technique requires no mask preparation, which increases flexibility, speeds up the manufacturing method for CNT-based sensors, and therefore lowers the fabrication costs [35]. Additionally, multilayers can be printed on top of each other with high accuracy and great ease. In this work, the single-nozzle drop-on-demand (DoD) inkjet printing technique is used for patterning the MWCNT aqueous dispersion as a planar coil. The ink formulation is prepared in such a way as to achieve a stable clog-free printing process, and patterns with different design parameters are printed. After the ink preparation, an optimization of the printing parameters to accomplish a high-accuracy printing pattern is conducted. Next, an impedance characterization of the printed planar MWCNT coil is determined to deepen the understanding of the printed pattern under applied strain force; then, the appropriate impedance model is extracted, fitted, and analyzed. Afterward, a wireless measurement technique using an impedance spectroscopy based on inductive coupling is established, and the change in the resonance frequency and the bandwidth of the printed CNT film is monitored.

## 2. Materials and Methods

The used MWCNTs were supplied by Southwest Nano Technology with a purity greater than 95%, outer diameters of 6–9 nm, and lengths less than 1 μm (as mentioned by the supplier). The MWCNTs were used without further purification or chemical treatment. The SDS powder with a purity ≥98.5% was purchased from Sigma Aldrich (Taufkirchen, Germany). All the dispersion experiments were carried out with 18 MΩ/cm deionized water. The aqueous suspension was prepared by dispersing 0.1 wt% MWCNTs in 0.5 wt% SDS surfactant. It was previously shown in our study [36] that the best dispersion quality is achieved at concentrations slightly above the Critical Micelle Concentration (CMC) value of the surfactant; this is due to the fact that when there is a non-uniform micelle formation of SDS at high concentrations, the dispersion quality starts to decrease. The suspension was then treated by a horn sonicator equipment (GM 3200 from Bandelin, Berlin in Germany) for 30 min in an ice bath to avoid overheating. After the dispersion had been accomplished, a centrifugation process using Sigma 2-16 PK equipment (Sigma Laborzentrifugen GmbH, Osterode am Harz Germany) took place at 3500 rpm for 45 min to precipitate the high CNT bundles. The preparation parameters, such as the sonication time, centrifugation time, and centrifugation force, were chosen based on previous intensive investigation in order to obtain the highest dispersion quality [37,38]. Only 80% of the upper part of the aqueous dispersion i.e., supernatant, which includes the unbundled CNTs, was used as ink for the inkjet printer. Before the ink was loaded into the inkjet printer, it was filtered via a 5 μm nylon mesh filter to reduce the possibility of nozzle blockage caused by nanotube clustering at the nozzle edge, which could cause a deviation in droplet trajectory. The printer nozzle has an orifice of 69 μm. A schematic of the dispersion preparation steps is illustrated in Figure 1.

Mainly, two sets of square planar coils having different fill factors ($\Theta_{fill}$), different inner diameters ($D_{in}$), different outer diameters ($D_{out}$), and different winding numbers (for instance 3, 4, and 5 windings) were printed on a pre-treated PET flexible polymer substrate with a thickness of 100 μm using a single-nozzle drop-on-demand (DoD) inkjet printer Autodrop System MD-P-801 from Microdrop technologies (Norderstedt, Germany). As is depicted in Figure 2, the first set has a $D_{in}$ constant at 8 mm and patterns with 3, 4, and 5 winding. In the second set, the $D_{out}$ is kept constant at 30 mm, and coils with different windings, mainly; 3, 4, and 5 winding were printed.

To avoid the formation of satellite droplets, the printing parameters of the printer were optimized; for instance, the pulse duration, pulse height, substrate temperature, print speed, and drop distance were adjusted to be 19 ± 2 μs, 80 ± 5V, 80 ${\;}^{\circ }$C, 8 mm/s, and 80 μm, respectively. To avoid any unusual jetting conditions, such as perturbations from the surrounding environment and diversion of the drop trajectory, the distance at which the droplet was released to land on the substrate was also optimized. The setup of the printing system and a snapshot of the drop formed using the optimized parameters are shown in Figure 3. Different numbers of layers (5, 10, 15, and 20 layers) were printed, and to ensure high reproducibility, a minimum of 3 coils from each pattern were printed and tested. However, to enhance electrical conductivity and facilitate the evaporation of the majority of surfactant residues, the printed coils underwent a drying process in a static oven set at 100 ${\;}^{\circ }$C for 7 min. The printed MWCNT planar coils were loaded in a tensile-compressive cyclic machine, and the readout mechanism was conducted by an impedance analyzer Agilent 4294A from Keysight Technologies Deutschland GmbH (Böblingen, Germany), which applies two 90° out-of-phase monochromatic small AC signals and measures the corresponding current response; thus, the complex impedance Z(ω) of the MWCNT printed planar coil can be calculated using the following:(1)$$\begin{array}{c} Z\left(\omega \right)=\frac{E_{0}\sin\left(\omega t\right)}{I_{0}\sin\left(\omega t-\Phi \right)} \end{array}$$
where $E_{0}$ and $I_{0}$ are the amplitude of voltage and current, respectively, ω = 2πf is the radial frequency in rad/s, *f* is a frequency in Hz, and Φ is the phase shift to the applied voltage.

The applied strain force was changed from 0 to 300 N with steps of 50 N, and the scanned frequency spectrum of the impedance analyzer was set from 40 Hz to 110 MHz. To ensure a linear interdependency between the applied voltage and the current response, the amplitude of the sinusoidal potential modulation of the impedance analyzer was set at 10 mV. The altering in the corresponding planar coil characteristics in terms of resistance and capacitance as a function of the strain was then analyzed. Furthermore, the influence of the design parameters, namely, the number of windings and the fill factor ($D_{in}$ and $D_{out}$) on the strain sensitivity and the change tendency of the printed planar coil, were investigated. To interrogate the wireless response of the printed MWCNT coils, an impedance analyzer was coupled with a reference spiral coil that resonates at 27.40 MHz and was then brought into the near vicinity (about 1 cm) of the printed MWCNT coil. When the current generated from the impedance analyzer passes through the spiral coil, it generates a magnetic field around it (Faraday’s law). When the MWCNT printed coil crosses this generated magnetic field, and based on the inductive coupling, a corresponding potential (voltage) drop and a current are formed in it; thus, the characteristics of the printed coil are determined. Once the printed coil undergoes a strain, the change in the MWCNT printed coil characteristic, such as the shift in the frequency, bandwidth, and magnitude at its initial resonance frequency, was determined, and the measurement setup of the inductive coupling is depicted in Figure 4.

## 3. Results

The influence of the number of printed layers on the electrical resistance of the printed planar coils having $D_{out}$ = 30 mm and $D_{in}$ = 12 mm are illustrated in Figure 5. It is observed as expected that, as the number of the printed layers increases, the resistance of the film decreases, for instance from 9.73, 4.53, 3.64, and 1.97 MΩ for 5, 10, 15, and 20 layers, respectively. This is referred to as the increase in the conductive paths formed by the CNTs between the end sides of the coils and therefore the formation of a 3D dense network that lowers the whole resistance of the thin film [7,11,12,14,39].

Besides an increase in the conductivity of the multilayered printed coils, a decrease in the printed coil’s standard deviation is noticed. For 5, 10, 15, and 20 layers, the standard deviation in the resistance is 0.53, 0.43, 0.25, and 0.02%, which is a sign of good reproducibility of the printed coils. The lowest resistance value is for the 20 printed layers, and thus, all the measurements are carried out on planar printed coils with 20 layers. Figure 6 displays a photographic, microscopic, and morphological picture of the printed planar coil with five windings, $D_{out}$ = 26 mm, and $D_{in}$ = 8 mm. It is revealed from Figure 6b that the quality of the printed layers is slightly affected by the formation of the coffee ring effect, which is due to the evaporation of the surfactant, caused by the interplay of ink viscosity and solute transport via solvent motion also because the CNTs tend to reorder themselves to the perimeter of the droplet due to an internal flux [32,33,34].

### 3.1. Stand-Alone Coil Characterization

The typical complex impedance (Z(ω)) response of the printed MWCNT planar coil and its equivalent circuit is illustrated in Figure 7a. From the Nyquist plot in Figure 7b, Randles’s circuit model is proposed for the randomly distributed network based on CNT, where $R_{t}$ is the tunneling resistance between the CNT; it is also observed from the Nyquist plot that it is not perfectly semicircle; therefore, a constant phase element (CPE) is used instead of a pure capacitor. This behavior is expected due to the non-homogeneity distribution of the MWCNT (Figure 6b), which will consequently result in a non-uniform distribution of the generated current within the structure due to the variations in its thickness. However, the CPE can be mathematically calculated as follows [40]:(2)$$\begin{array}{c} Z\left(\omega \right)=Q^{-1}{\left(j\omega \right)}^{-n} \end{array}$$
where 0 < *n* < 1, and *Q* is a constant with units (siemens.second). For *n* = 1, the CPE is a perfect capacitor, and thus, *Q* equals the capacitance.

Based on that, the value of the true capacitance ($C_{true}$) can be also calculated using the following equation [40]: (3)$$\begin{array}{c} C_{true}=Q{\left(\omega_{max}\right)}^{n-1} \end{array}$$
where $C_{true}$ is the true capacitance, and $\omega_{max}$ is the frequency at which the imaginary component reaches the maximum.

The quantum capacitance per unit length $C_{CNT}$ can be estimated by [41]
(4)$$\begin{array}{c} C_{CNT}=L\times \frac{2e^{2}}{\hbar V_{F}} \end{array}$$
where *L* is the CNT length, *e* is the electron charge, *h* is Planck’s constant, and $V_{F}$ is the Fermi velocity, which is equal to $8\times 10^{5}$ m/s for metallic CNT.

The tunneling resistance $R_{CNT}$ between the neighboring CNTs can be estimated using Simon’s formula [42].
(5)$$\begin{array}{c} R_{tunneling}=\frac{V}{AJ}=\frac{h^{2}\times d}{A\times e^{2}\times \sqrt{2m\lambda }}exp\left(\frac{4\pi d}{h}\sqrt{2m\lambda }\right) \end{array}$$
where *V* is the potential difference, *J* is the tunneling density, *A* is the cross-sectional area of the filler, *h* is Planck’s constant, *d* is the distance between filers, *e* is the electron charge, *m* is the electron mass and λ is the height of the electrical barrier.

However, the total contribution of the quantum capacitance to the total capacitance of the thin films are negligible [43,44]; thus, it can be ignored, and consequently, the tunneling (tube-tube) resistance dominates. Therefore, only the tunneling effects appear on the EIS spectrum [45]. Under applied strain force, a shift in the Nyquist plot occurs, and by extracting the parameters at each applied force, those parameters can be derived (Figure 8). A relationship between the strain and the change in the electrical resistance (ΔR/$R_{0}$) of the printed planar structure with different fill factors is plotted in Figure 9. It is obvious that as the applied strain force increases, the resistance of the printed coil structure increases; this is referred to as the change in the tunneling distance between the MWCNTs and is illustrated in Figure 8.

For printed coils with $D_{out}$ = 30 mm, a linear piezoresistive behavior is noticed, and increasing the number of windings increases the sensitivity. And, for printed coils with $D_{in}$ = 8 mm, an exponential trend can be visualized and tends to become more linear for a high number of winding, but the sensitivity tends to fall with a higher fill factor. The increase in the linearity for a higher number of windings is attributed to the increase in the fill factor of the planar coil. The piezoresistive sensitivity of the printed planar coils-based MWCNTs is evaluated by mean of gauge factor (*k*),
(6)$$\begin{array}{c} k=\frac{\Delta R/R_{0}}{\epsilon } \end{array}$$
where $R_{0}$ is the initial resistance before loading, ΔR is the change of the resistance, and ϵ is the strain ϵ = δ$L/L_{0}$.

As the maximum applied force is 300N, the resulting strain is 1.6%. For printed planar coils with $D_{out}$ = 30 mm, a linear behavior is noticed with gauge factor 4.6, 24.9, and 29 for three, four, and five windings, respectively. On the other hand, an exponential trend is seen in printed coils with a reduced fill factor, denoted as $D_{in}$ = 8 mm. As a result, two separate gauge factors are observable. As previously explained, it is appealing that there is a correlation between the critical strain value and the number of windings. Table 1 summarizes the values for the k-factor for $D_{in}$ = 8 mm and its related critical strain value. The printed structures also exhibited good hysteresis behavior for $D_{out}$ = 30 mm, and the change in the normalized resistance before and after loading for the coils are 7.9%, 7.6%, and 3.7% for three, four, and five windings, respectively. For $D_{in}$ = 8 mm, almost no change in the normalized resistance before and after loading is noticed.

For the CPE element, no clear trend change was noticed, and also, almost no change in the true capacitance was remarked (change is less than 0.002%) [40,45]. This is referred to by the very low tunneling capacitance value that appears to not be detectable by the EIS measurement. The piezoresistive behavior of the printed planar MWCNT coils can be explained as follows: before the load is applied, all CNTs are randomly distributed onto the substrate, as demonstrated in Figure 8b, and thus, the MWCNTs inside the matrix are overlapping at the contact locations, rather than being arranged in an end-to-end configuration. However, an alignment in the direction of the applied strain force occurs as the coils undergo a load; this alignment changes the contact between CNTs, which clues to the appearance of the tunneling effect as the main effect [11,12,22,46].

### 3.2. Inductive Coupling Characterization

When the reader coil is brought in close proximity to the printed planar MWCNT coils, a shift in the resonance frequency to a new frequency called the coupling resonance frequency will occur due to the mismatch in the resonance frequencies of both coils, which will also affect the energy transfer efficiency between the coupled coils. Figure 10 shows the displacement in the coupling frequency for the primary reader coil resonating at 27.4 MHz and five winding printed planar MWCNT coils with $D_{in}$ = 8 mm. Typically, for two asymmetrical pairs of coupled coils, the coupling resonance frequency $f_{r}\left(coupling\right)$ is calculated as follows: (7)$$\begin{array}{c} f_{r}\left(coupling\right)=\frac{1}{2\pi }\sqrt{\frac{L_{m}}{\left(L_{1}\times L_{2}-{L_{m}}^{2}\right)\times C_{m}}} \end{array}$$
where $L_{m}$ is the mutual inductance, $L_{1}$ and $L_{2}$ is the internal inductance of the reader coil and printed coil, respectively, and $C_{m}$ is the mutual capacitance.

Under applied force, no further shift in the frequency is noticed, and the only change was in the amplitude i.e., the bandwidth. This is referred to by the very low change in capacitance of the secondary coils (printed planar coils), which therefore did not affect its resonance frequency, as can be noticed in the following equation: (8)$$\begin{array}{c} F_{r}=\frac{1}{2\pi \sqrt{LC}} \end{array}$$
where $F_{r}$ is the resonance frequency of printed coils, L and C are the inductance, and the true capacitance of the coil.

The change in amplitude for all patterns is illustrated in Figure 11. Coils with a lower fill factor, i.e., $D_{in}$ = 8 mm, show higher sensitivity and better linearity than coils with $D_{out}$ = 30 mm. Independent of the design parameter, as sensitivity increases for a higher number of windings, this can be attributed to the higher energy transfer that occurs between the coils.

## 4. Conclusions

In this article, the MWCNT is dispersed in an anionic SDS surfactant and used as an ink to print planar coils, which are used to determine strain wirelessly. Inkjet printing offers many advantages such as high reproducibility, fast deposition, and thickness control, as compared to other deposition methods such as screen printing, electrophoresis, physical liquid deposition and layer-by-layer deposition. Increasing the number of printing layers reduces the resistance of the whole pattern because the MWCNTs form more conducting paths. The results of the stand-alone coil characterization showed that the change in the tunneling resistance of the MWCNT coil was proportional to the strain, and no influence on the tunneling capacitance was noticed. The coupled antenna coil was also examined, and the findings indicated that the amplitude, which represents the variation in bandwidth, exerts the greatest influence under applied strain force in comparison to the resonance frequency.

The amplitude increased almost linearly with the strain and the number of windings. As more windings were introduced to the printed coils, it was clearly observed that there is a distinct trade-off in how linearly the strain responds. Whereas, with an increase in the number of windings, the sensitivity to strain consistently rises.

## Figures and Tables

**Figure 1 sensors-24-01585-f001:**
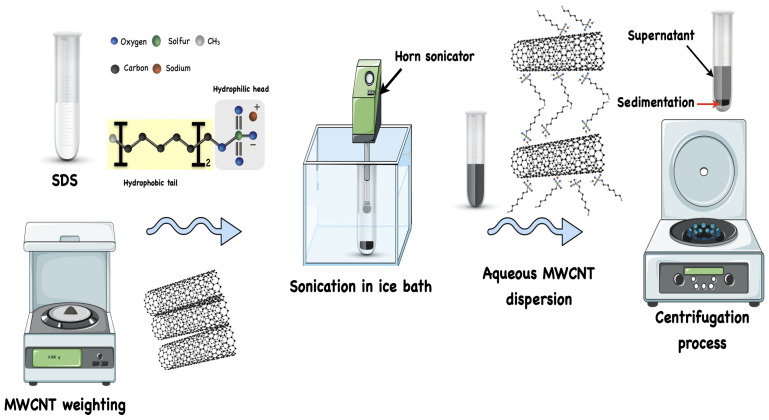
Schematic illustration of MWCNT aqueous dispersion preparation.

**Figure 2 sensors-24-01585-f002:**
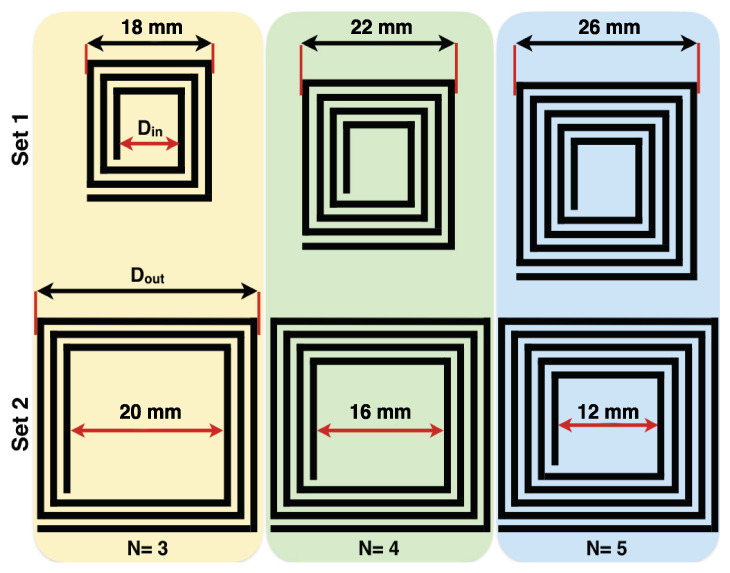
Different patterns of the printed planar coil-based MWCNTs dispersion. (Dimensions are in mm, N denotes the number of windings), Set 1: $D_{in}$ = 8 mm, and Set 2: $D_{out}$ = 30 mm.

**Figure 3 sensors-24-01585-f003:**
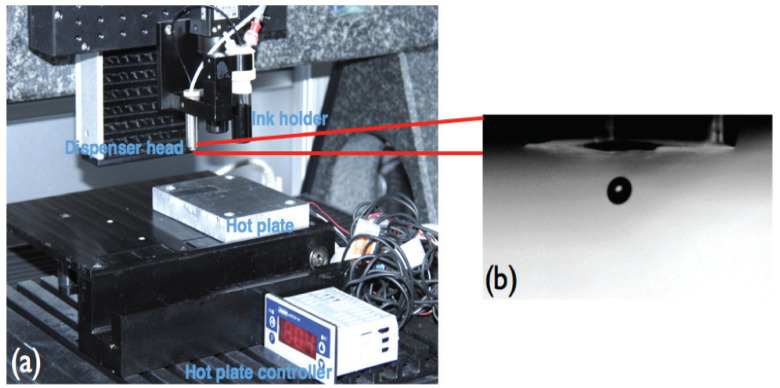
Optimization and Drop Formation Analysis using an inkjet printer: (**a**) single nozzle inkjet printing system, (**b**) snapshot of a drop formed using the optimized parameters coming out of the dispenser head.

**Figure 4 sensors-24-01585-f004:**
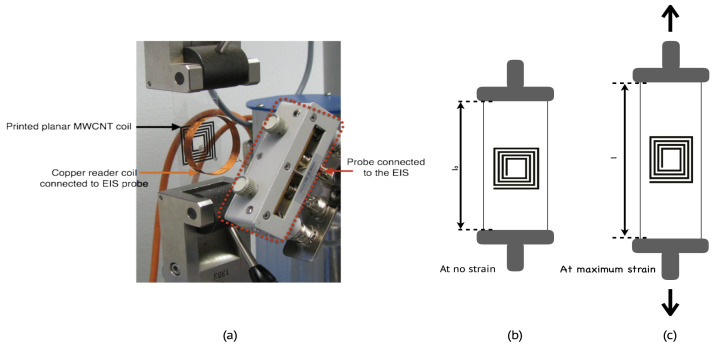
Measurement setup for the wireless strain measurement (**a**) coupling of the reader antenna with the printed CNT coil and substrate at (**b**) no applied strain force, and (**c**) at maximum applied strain force.

**Figure 5 sensors-24-01585-f005:**
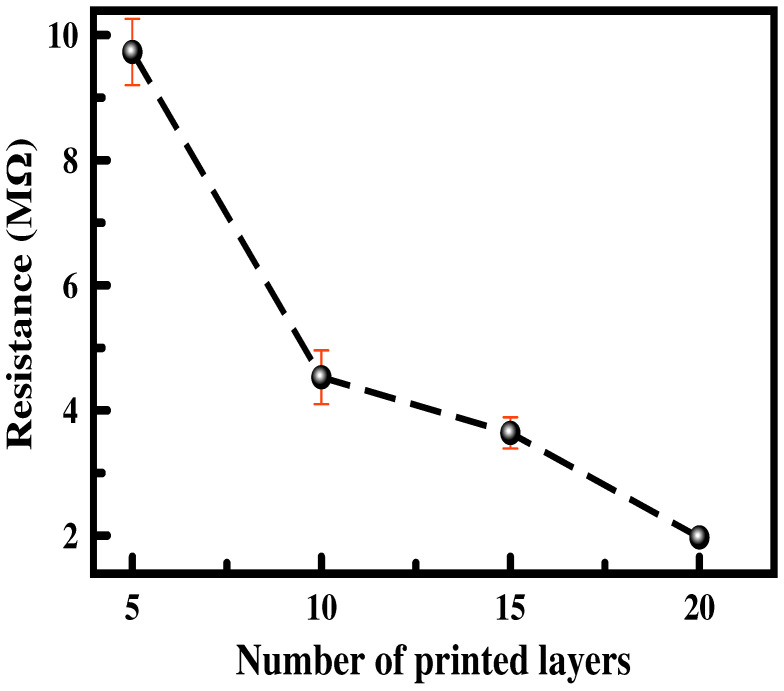
Change in printed coil resistance with 5 windings having $D_{out}$ = 30 mm and $D_{in}$ = 12 mm for different numbers of printed layers.

**Figure 6 sensors-24-01585-f006:**
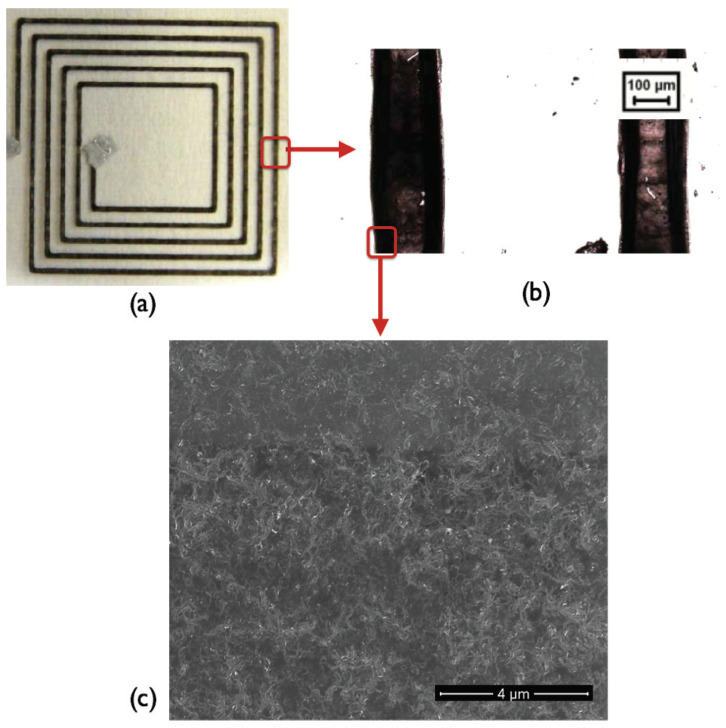
Printed coil with 5 windings, $D_{out}$ = 26 mm and $D_{in}$ = 8 mm (**a**) Photographic inset, (**b**) Microscopic snapshot, (**c**) SEM of MWCNT distribution.

**Figure 7 sensors-24-01585-f007:**
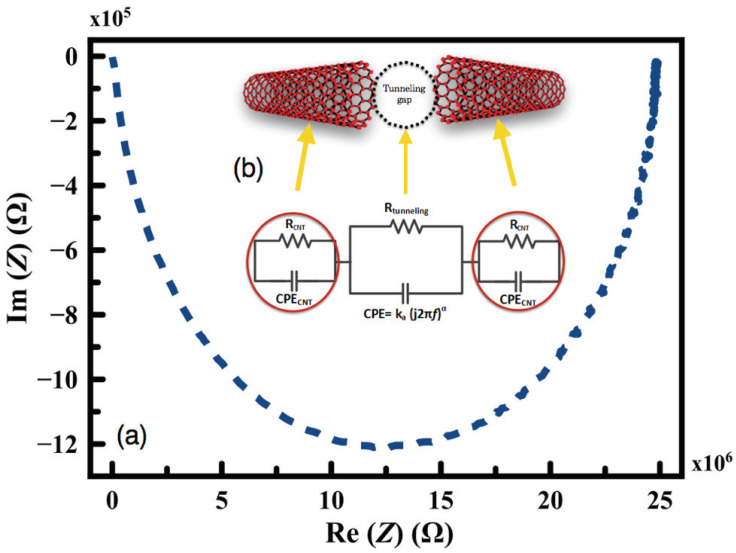
EIS Characterization and Modeling of Printed MWCNT Coils: (**a**) impedance spectrum of the printed MWCNT coil, (**b**) equivalent representative circuit of the printed coils.

**Figure 8 sensors-24-01585-f008:**
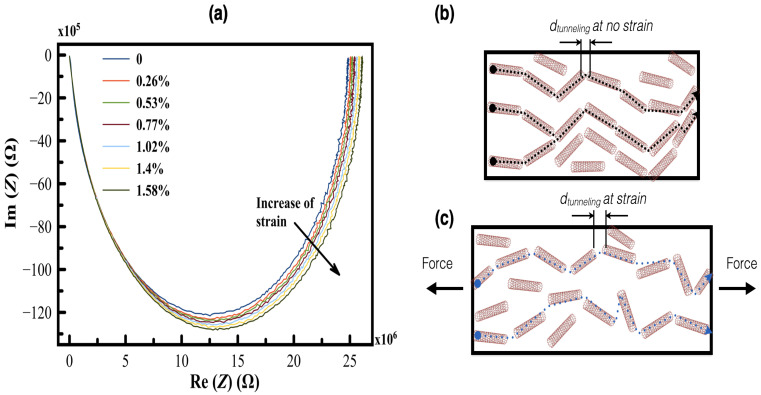
Impedance Spectrum and schematic illustration of the printed coils under applied strain force: (**a**) shift in the impedance spectrum as a function of the strain, (**b**,**c**) schematic description of the change in the tunneling distance before (black-doted line) and under strain (blue-doted line).

**Figure 9 sensors-24-01585-f009:**
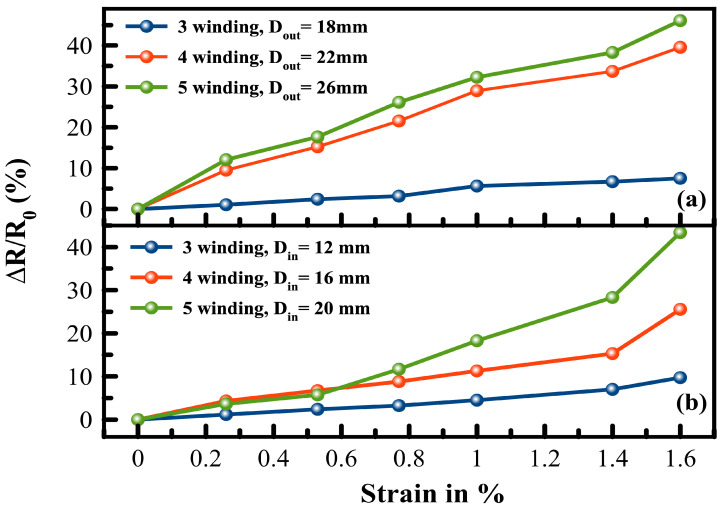
Change in the printed coil resistance as a function of the applied strain force, (**a**) coils with $D_{out}$ = 30 mm, and (**b**) coils with $D_{in}$ = 8 mm.

**Figure 10 sensors-24-01585-f010:**
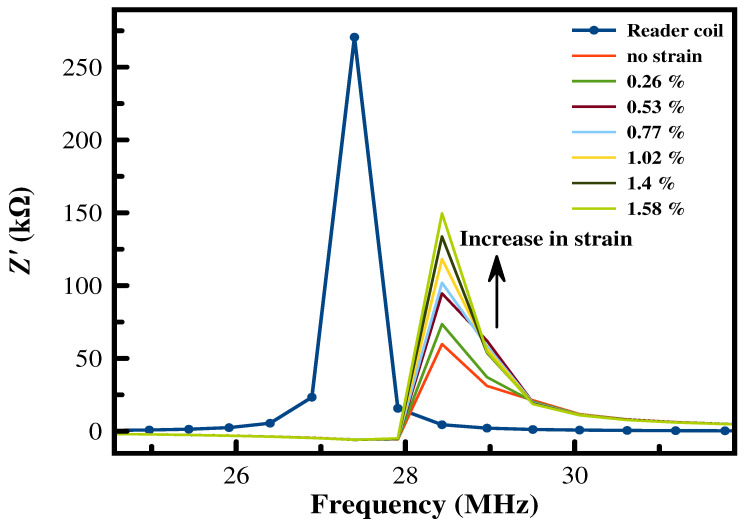
Inductive coupling between printed MWCNT planar coils and reader coils under strain for five winding coils with $D_{out}$ = 26 mm and $D_{in}$ = 8 mm at 1 cm reading distance.

**Figure 11 sensors-24-01585-f011:**
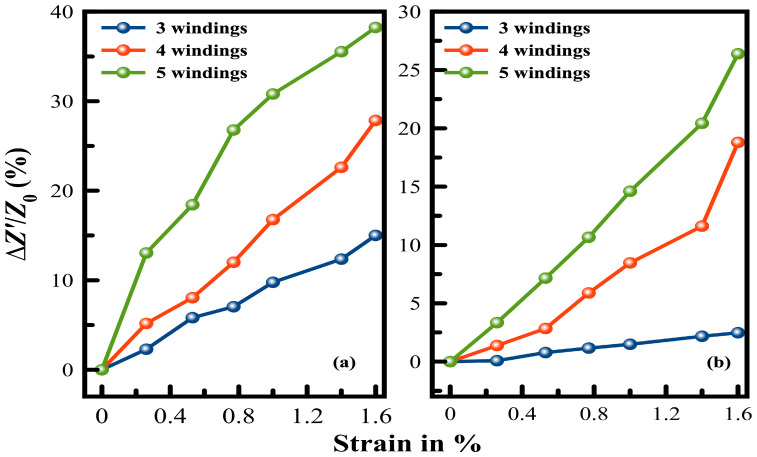
Change in the bandwidth as a function of the strain for patterns with (**a**) coils with $D_{in}$ = 8 mm, and (**b**) coils with $D_{out}$ = 30 mm at 1 cm reading distance.

**Table 1 sensors-24-01585-t001:** The strain sensitivity of the printed planar coils based on MWCNTs having 20 printed layers and $D_{in}$ = 8 mm.

Number of Winding	Strain	k-Factor
3 Windings	≤0.8%	14.6
	≥0.8%	17.5
4 Windings	≤1.3%	11.46
	≥1.3%	14.66
5 Windings	1.6%	5.49

## Data Availability

Data are contained within the article.

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
