# Peer review of "Inkjet-Printed Multiwalled Carbon Nanotube Dispersion as Wireless Passive Strain Sensor"

_sensors, 2024, doi:10.3390/s24051585_

Round 1
Reviewer 1 Report
Comments and Suggestions for Authors
See the attached file.

See the attached file.
Author Response
We want to sincerely thank the reviewer for their insightful feedback, which has significantly raised the quality of our work. Their enlightening comments have been crucial in helping us polish our work and maintain its academic integrity.

Reviewer 2 Report
Comments and Suggestions for Authors
First of all, congratulations on your excellent work. I would appreciate that you take into account my suggestions.
- Line 99. This sentence needs the image (Figure 2) to be supported. On the paper, it happens sometimes that the image that you are referring to is so far inside the text that it could be an inconvenience to follow the development of your research. Please consider relocating the images close to the text part where they are mentioned.
- Figures 8 and 9 are too far away from their mention. Also, a table mentioned after them is placed before them. It is difficult to understand. As in the previous comment,
- As it is said in Line 209, I understand that the hysteresis shown refers to one cycle. Did you test or study the effect on the resistance after more than one cycle? Durability is also a factor to take into account.
- Figure 10 caption. How can be differenced top graph from both graph? A graph refers to Dout=30, but the top graph refers to a different Dout. (a) and (b) references should be marked on the graph.
Comments on the Quality of English LanguageA review of the English language is needed along with the paper. Some examples that I could find during the review.
- Line 139. The sentence should be rewriten.
-Line141-143. The understanding of the idea could be improved by changing the way of explaining it.
- Line 247. ''In this journal...'' journal could be substituted by paper, work, or research to individualize your work.
- Line 249. Reproducibili. Reproducibility
Author Response

(The authors gave the same response as above.)

Round 2
Reviewer 1 Report
Comments and Suggestions for Authors
The authors responded correctly to my comments
I hope that their next study gives more space for more experimental work such as repeatability and long time stability.